# Post-Chemoradiation Metastatic, Persistent and Resistant Nodes in Locally Advanced Rectal Cancer: Metrics and Their Impact on Long-Term Outcome

**DOI:** 10.3390/cancers15184591

**Published:** 2023-09-15

**Authors:** Felipe A. Calvo, María Tudela, Javier Serrano, Mercedes Muñoz-Fernández, María Isabel Peligros, Pilar Garcia-Alfonso, Emilio del Valle

**Affiliations:** 1Hospital General Universitario Gregorio Marañón, 28007 Madrid, Spain; maria.tudela@salud.madrid.org (M.T.); mmunozfernandez2@salud.madrid.org (M.M.-F.); isabel.peligros@salud.madrid.org (M.I.P.); pgalfonso@salud.madrid.org (P.G.-A.); emilio.valle@salud.madrid.org (E.d.V.); 2Department of Oncology, Clinica Universidad de Navarra, 28027 Madrid, Spain; fserranoa@unav.es

**Keywords:** post-chemoradiation metastatic, locally advanced rectal cancer, neoadjuvant therapy, intraoperative electrons, nodal metastases, chemotherapy

## Abstract

**Simple Summary:**

The tumor response to neoadjuvant radiochemotherapy is one of the most important prognostic factors in patients with locally advanced rectal cancer. Locally advanced rectal cancer contains migrant cell populations, the source of locoregional metastases. Neoadjuvant radiochemotherapy treatment identifies with pathological certainty lymph node metastatic radiochemoresistance. These patients have a higher risk biology for regional and systemic migrant recurrence that requires consideration of an evidence-based change in therapeutic strategy. The aim of this work is to analyze the impact of post-neoadjuvant nodal status as an independent prognostic factor in locally advanced rectal cancer to implement adjuvant chemotherapy administration tailored to each patient’s individualized risk response. The hypotheses that we evaluate in this study can guide the individualization of the therapeutic approach in patients with post-neoadjuvant metastatic lymph node persistent rectal cancer, as well as the definition of subgroups based on certain patient characteristics and tumor biology.

**Abstract:**

Background: The purpose of this study was to evaluate the long-term oncological progression pattern of locally advanced rectal cancer patients with post-neoadjuvant nodal metastatic disease (ypN+) and correlate potential prognostic features associated with proven radiochemoresistant nodal biology. Methods: Individual patient data (100 variables) from a 20-year consecutive single-institution multidisciplinary experience (1995–2015), delivering multimodal therapy to rectal cancer patient candidates for radical treatment, including a neoadjuvant component and surgical resection with or without intraoperative radiotherapy followed by optional adjuvant chemotherapy. The ypN+ disease data was registered in the context of initial staging categories post-neoadjuvant T status (ypT). Results: Data on 487 patients showed histologically confirmed diagnoses of metastatic nodal disease in 108 specimens (ypN+, 22.1). There was a significant age difference (*p =* 0.009) between the ypN groups: age ≥ 65 was 57.6% in pN0 and 43.5% in ypN+ and patients aged < 65 constituted 42.4% of pN0 and 56.5% of ypN+. According to the clinical stage there were statistically significant differences (*p =* 0.001) in the categories’ distribution: ypN+ patients 10.8% were stage II and 89.2% were stage III. Univariant analysis on outcome variables showed statistically significant differences in overall survival at 7 years (63.8% vs. 55.7%, *p* = 0.016) disease-free survival (DFS) (78% vs. 53.8%, *p* = 0.000) and local recurrence-free survival (LRFS) (93.6% vs. 84%, *p =* 0.002). Conclusions: The presence of nodal metastases (ypN+) after neoadjuvant therapy containing long-course pelvic irradiation severely impacts the long-term outcome for patients with locally advanced rectal cancer and correlates with multiple clinical and therapeutic variable metrics. Implementation of local and systemic therapies should be adapted and intensified in relation to the finding of ypN+ category in surgical specimens.

## 1. Introduction

The tumor response to neoadjuvant radiochemotherapy is one of the most important prognostic factors in patients with locally advanced rectal cancer. The clinical and histological response is higher when the cellular grade (G) is lower. The lower the T and N stage, the higher the radiation dose and the longer the interval between neoadjuvant and surgery (6–8 weeks); a total of 80% of patients with pathological complete response (pCR) undergo surgery within 60 days after the end of irradiation [1].

Locally advanced rectal cancer contains migrant cell populations, the source of locoregional metastases. Neoadjuvant radiochemotherapy treatment identifies with pathological certainty lymph node metastatic radiochemoresistance. These patients have a higher risk biology for regional and systemic migrant recurrence that requires consideration of an evidence-based change in therapeutic strategy. There are prognostic factors that may guide individualization of the therapeutic approach in patients with post-neoadjuvant nodal persistent metastatic rectal cancer (ypN+). This pathological subcategory is associated with adverse prognosis in long-term survival [2].

The aim of this work is to analyze the impact of post-neoadjuvant nodal status as an independent prognostic factor in locally advanced rectal cancer in an attempt to implement adjuvant chemotherapy administration tailored to each patient’s individualized risk response. We evaluate hypotheses of possible prognostic factors that can guide the individualization of the therapeutic approach in patients with post-neoadjuvant metastatic lymph node persistent rectal cancer (ypN+), as well as the definition of subgroups based on certain patient characteristics and tumor biology that can guide us in making therapeutic decisions regarding the intensity and selection of anti-neoplastic agents.

## 2. Materials and Methods

To evaluate the oncological progression pattern of post-neoadjuvant ypN+ patients, potential prognostic factors related to radiochemoresistant rectal cancer are analyzed. The risk of oncological progression is categorized in relation to individual clinical and therapeutic data. Therapeutic effect is correlated with progressive clinical outcomes and oncological prognostic categories of interest.

The setting of the study is institutional: Hospital General Universitario Gregorio Marañón (Madrid, Spain). Methodologically it is an observational, analytical–descriptive, epidemiological study, with prospective registration of data in a consecutive controlled cohort of patients with locally advanced rectal cancer treated with neoadjuvant radiochemotherapy followed by surgery and occasional posterior hemipelvis boost irradiation during the procedure. The study period spans 20 years, from April 1995 to March 2015.

The target population of the study included 488 patients with a histologically confirmed diagnosis of locally advanced rectal adenocarcinoma (cT3-4 N0 or N+ M0) referred to the Hospital General Universitario Gregorio Marañón (Department of General and Digestive Surgery or Department of Oncology). The inclusion criteria was patients who were candidate for radical treatment with a neoadjuvant component and surgical resection with or without intraoperative radiotherapy followed by adjuvant chemotherapy if required by non-randomized consecutive sampling. The definitive pathology report of the surgical specimen should be completed. The database includes 100 variables of diagnostic, clinical, pathological and evolutive items of interest per patient. The external beam radiotherapy technique was 3D conformal irradiation in all patients. Neoadjuvant induction oxaliplatin-based chemotherapy was introduced in practice on 05/1998 upon an institutional intramural research program. Adjuvant postoperative chemotherapy practice was implemented with oxaliplatin-based regimes in 2000 for patients with high-risk pathological findings for recurrence (high ypT and/or ypn+ categories).

All the participants included in the study were assessed by physical and clinical examination, colonoscopy, imaging study and biopsy with confirmation histology. The clinical assessment of the lymph node involvement of the disease was carried out with different tests: 44.2% were diagnosed by MRI, 32.9% by endorectal ultrasound, 22.3% by CT and only 6% were diagnosed with lymph node involvement by PET/CT. No significant differences were detected between the two study groups. Data was recorded on the tables assuming the non-available information in the following variables: quality of mesorectum (15.1%), ASA category (51.9%), clinical node staging (cNx status) (21%), unknown lymphovascular (29.8%) or perineural invasion (30%).

The program used to carry out the statistical analysis of the data is SPSS 20.0 for Windows (version 20.0.0). The statistical analysis consists of a descriptive part detailing the characteristics and distribution of the different variables included in the database (socio-demographic, diagnoses, treatments carried out, complications derived from these, recurrence and its treatment and the patient’s current situation).

For quantitative variables, normality was verified using the Shapiro–Wilks test for samples of less than 50 patients or the Kolmogorov–Smirnov test for samples of more than 50 patients. Quantitative variables with normal distribution were presented as mean and standard deviation, and those without as median and interquartile range (P75-P25). Comparison of independent groups was performed using Student’s *t*-test if the dependent variable followed a normal distribution, and using the Mann–Whitney U-test in case of non-normal distribution. For qualitative variables, results were presented as absolute counts and percentages. The association between two dichotomous qualitative variables was studied using the Chi-square test if the expected frequency in all cells was not less than five and otherwise using Fisher’s exact test. For polychotomous variables, only the Chi-square test was used. Subsequently, a univariable logistic regression analysis of clinical, pathological, and therapeutic factors that could influence the occurrence of local or distant recurrence or survival was performed. Variables that showed significant differences in the univariable analysis were analyzed using the Cox multiple-factor analysis method. Survival probability was calculated using the Kaplan–Meier method. Comparison of survival between groups according to different prognostic factors was performed using the log-rank test. 

Overall survival, disease-free survival and local and distant recurrence-free intervals were computed from the date of surgical treatment. Patients who were alive without disease at the time of study completion or those who died without evidence of tumor recurrence were considered censored. Relapse-free survival and overall survival were estimated at 7 years to explore the potential incidence of late relapses in the context of proven metastatic nodal cancer biology.

To study the association of the probability of survival with different qualitative and quantitative predictor variables, the Cox proportional hazards model was used, first univariable and then multivariable. The Cox model relates the relative risk of death or the occurrence of an event to the different selected prognostic factors and calculates the risk ratio (hazard ratio) and the 95% confidence interval for each of these variables. 

This study was conducted in accordance with the standards of good clinical practice with full acceptance of the ethical standards in force (Declaration of Helsinki, Brazil 2013 revision) and respecting all aspects established in the current legislation on clinical research. This study is limited by the methodological deficits inherent in descriptive, observational, non-randomized analyses of an institutional cohort.

## 3. Results

Descriptive statistical analysis examines data from 488 patients with a histologically confirmed diagnosis of locally advanced adenocarcinoma (ADC) of the rectum treated from April 1995 to March 2015.

### 3.1. Demographic Characteristics

The sex distribution was 60.9% male and 39.1% female, distributed homogeneously in both groups (*p* = 0.699). The mean age was 64 years (sd +/− 12 years) so that 45.5% were <65 years and 54.5% were ≥65 years. There was a significant age difference (*p* = 0.009) between the two groups, thus, age ≥ 65 was 57.6% in pN0 and 43.5% in ypN+ and patients aged <65 constituted 42.4% of pN0 and 56.5% of ypN+. 

Regarding the functional and general condition of the patient classified into two groups, 79.8% were low ASA (ASA I-II) and 20.2% high ASA (ASA III-IV) with no differences between the ypN0 and ypN+ groups (*p* = 0.057). When considering the existence of previous abdominal surgery, 73.7% had no previous abdominal surgery and 26.3% had another previous surgical procedure, with no statistically significant differences between the two groups (*p* = 0.778) (Table 1).

The clinical tumor stage (cT) was similar in both groups with no significant differences between them (*p* = 0.075). Of the entire series, 5.9% were cT2, 74.6% cT3, 17.2% cT4 and only 2.3% cTx. In clinical nodal stage (cN), significant differences (*p* = 0.001) were observed between the two groups. Among the ypN0 patients, 26.8% were cN0, 69.2% cN+ and 3.9% cNx. In the ypN+ group, 10.2% were cN0, 84.3% cN+ and 5.6% cNx. 

The distribution of the series according to clinical stage was in favor of stage III with 75.8% of all patients and 24.2% for stage II. When comparing both groups there were statistically significant differences (*p* = 0.001), so that in the ypN0 group 27.9% of the patients belonged to stage II and 72.1% were stage III, on the other hand, among the ypN+ patients 10.8% were stage II and 89.2% were stage III (Table 2). 

Regarding the chemotherapy regimens received in the context of neoadjuvant therapy, 69.8% received induction with oxaliplatin and 30.2% did not. Regarding concomitant chemotherapy, 8% received continuous infusion 5FU, 2.5% Xelox, 18.3% oral tegafur and 62.2% oral oxaliplatin-tegafur [3]. Most patients completed neoadjuvant QT 79.3% and only 20.7% had to discontinue without completing it. There was no difference between the two groups (*p* = 0.664). Time from end of neoadjuvant to surgery was <8 weeks in 80.3% and ≥8 weeks in 19.7% of patients. External beam radiotherapy characteristics have been described previously [4]. The total dose ranged from 45 Gy to 50.4 Gy with daily fractions of 1.8 to 2 Gy and D conformal irradiation technique.

The surgical techniques employed are as follows: low anterior resection 41.3%, abdominoperineal amputation 29.7%, anterior resection 11.2%, ultra-low anterior resection 15.3% and pelvic exenteration 1.4%. There were no significant differences between the two groups. Total mesorectal excision (TME) was performed in 88.5% of all patients in the series. In the ypN0 group 88.2% and in the ypN+ group 89.8% (*p* = 0.634).

Intraoperative radiotherapy as a method of presacral boosting [5,6] was administered to 80.1% of all patients. In the ypN0 group 79.2% and in the ypN+ group 83.3% of patients (*p* = 0.34).

### 3.2. Complete Responders in T Category (ypT0)

The histological “T” stage distribution observed after analyzing the post-neoadjuvant surgical specimen (ypT), was: 59 patients (12%) with pathological T complete response were found, all of them were ypN0. In the ypN0 group 62.2% were ypT1, T2 and 35.8% ypT3, T4; on the other hand, in the ypN+ group 24.1% were classified as ypT1, T2 and 75.9% as ypT3, T4, with a statistically significant difference (*p* = 0.001) in the proportion of more advanced ypT stage tumors in the ypN+ group.

### 3.3. T-Downstaging

The assessment of tumor downstaging was distributed as follows:

In the ypN0 group, a favorable response to neoadjuvant treatment was observed in 71.4.1% of patients, with a 1T downstaging in 40.3%, a 2T downstaging in 15.5% and a 3T downstaging in 16.9%. However, in 25.5% of patients, the tumor did not respond to neoadjuvant treatment and therefore there was no downstaging; in 2.7% there was not only no response, but also a worsening of the pre-treatment classification (cT) due to previous understaging. In the ypN+ group, a favorable response to neoadjuvant therapy was observed in 36.5% of patients, of whom 1.7% experienced a 1T downstaging and 4.8% had a 2T downstaging. No patient had a 3T downstaging. However, 59.6% of the ypN+ patients were chemo-radioresistant, i.e., there was no downstaging. A total of 3.8% did not respond or worsened from pre-treatment classification (cT). 

### 3.4. N-Downstaging

In the assessment of nodal downstaging from initial clinical stage (cN) it was observed that of the 113 cN0 patients, cN+ 354 and cNx 21. In the pathological restaging 102 patients (20.9%) were ypN0, and 11 patients (2.25%) were ypN+ potentially understaged with imaging tests. Of the 354 cN+ patients, 263 patients (53.8%) were reclassified as ypN0 and therefore assumed to be responders. In 91 patients (18.6%) ypN+ status was ascertained and therefore interpreted as a group of non-responders. On the other hand, there were 21 cNx patients and of these 15 (3%) were ypN0 and 6 patients (1.2%) were ypN+. In the analyses of nodal downstaging by ypN, 28% of the ypN0 group did not experience any downstaging or improvement in nodal classification, it can be speculated that prior to neoadjuvant treatment they were cN0, while 72% of the ypN0 patients exhibit some degree of downstaging post-neoadjuvant. As for the ypN+ group 89.2% of patients did not experience any degree of nodal downstaging (non-responders). In addition, 10.8% had worsening, either patient who were cN0 and despite neoadjuvant treatment, pathology revealed that they were ypN+ patients and therefore clinically understaged. Significant differences were observed between the two groups (*p* = 0.001) (Figure 1).

### 3.5. Radial Circumferential Margin

In 98.4% of all patients the radial circumferential margin was >1mm, while it was found to be involved (<1 mm) in 1.6% of all study participants. The ypN0 patients had a free radial margin (>1 mm) in 99.2% and only 0.8% (three patients) were found to be affected. While in the ypN+ group 95.3% of patients had a free radial margin and 4.7% were affected (five patients). These differences were statistically significant (*p* < 0.01) because, although the absolute difference in radial circumferential margin involvement between the two groups is very small (3.9%), the sample size is large enough (478 patients) to reach significance.

### 3.6. Perineural and Lymphovascular Invasion

The presence of perineural invasion was found in 8.1% of ypN0 patients and in 48.7% of ypN+ patients. On the other hand, the presence of lymphovascular invasion was reported in 7.6% of ypN0 patients and 54.5% of ypN+ patients. These differences were statistically significant (*p* < 0.01) with a higher rate of perineural and lymphovascular invasion in ypN+ patients. This is because radioresistant tumors (ypN+) are usually aggressive tumor biology neoplasms and therefore have a greater presence of perineural and lymphovascular invasion (Table 3). 

### 3.7. Patterns of Progression

With a median follow up of 7 years (P25 48 months-P75 134.8 months) the following events occurred in the series: a total of 182 deaths, overall survival (OS) of 62% and a total of 131 recurrences of any type were diagnosed, giving an event free survival (EFS) for the entire series of 72.7%. 

The topographic pattern of recurrences was analyzed according to location; the results were as follows:Systemic recurrence (metastases only): seen in 90 patients; 58 (15.5%) in the ypN0 group and in 32 patients (30.2%) in the ypN+ group;Local recurrence (exclusively local): this was found in 16 patients, 9 (2.4%) in the ypN0 group and in 7 patients (6.6%) in the ypN+ group;Mixed relapse (local + systemic): 25 patients were targeted, 15 (4%) in the ypN0 group and in 10 patients (9.4%) in the ypN+ group. Statistically significant differences were observed between the two groups (*p* = 0.001), with a higher percentage of recurrences of any type in the ypN+ group;Local recurrence-free survival (LRFS): a total of 41 local recurrences (either only LR or local associated with metastases) were reported, giving an LRFS of 91.4%. In the ypN0 group there were 24 local recurrences (6.4%) and in the ypN+ group 17 patients (16%) had some form of LR (*p* = 0.002).

According to the location of the recurrence, the results were as follows:Anterior recurrence: There were 3 (0.8%) among ypN0 patients and 0 (0%) in ypN+ patients;Central recurrence: 11 (2.9%) were diagnosed among ypN0 patients and 3 (2.9%) among ypN+ patients;Lateral recurrence: There was 1 (0.3%) among ypN0 patients and 5 (4.8%) in ypN+ patients;Presacral recurrence: 9 (2.4%) were detected among ypN0 patients and 8 (7.6%) among ypN0 patients.

### 3.8. Univariable Analysis 

#### Overall Survival

Table 4 summarizes the univariable analysis for overall survival according to the main variables influencing overall survival. The variables that significantly influenced overall survival in the univariable analysis were as follows: age, anesthetic risk, tumor height (in the lower third), surgical technique, laparoscopic approach, existence of complications, type of resection R0 vs. R1, quality of the mesorectum, radial circumferential margin, degree of tumor regression, stage ypT, stage ypN, total number of positive nodes, percentage of affected nodes, positive node ratio ≥0.28, positive node ratio ≥0.5, existence of perineural and perivascular invasion and administration of adjuvant chemotherapy. Age <65 behaved as a protective factor for overall survival, improving survival by up to five fold (HR 0.48 [0.36–0.66]). The laparoscopic approach also behaved as a protective factor for overall survival, improving survival by almost four fold (HR 0.38 [0.23–0.64]). All other significant variables in the univariable analysis behaved as risk factors for overall survival: ASA 3–4 (HR 2.11 [1.15–3.88]); location in the lower third of the rectum (HR 1.49 [1.10–2.04]); the performance of a PPP (HR 1.75 [1.30–2.35]); the development of complications (HR 1.79 [1.31–2.44]); the R1 resection status (HR 5.13 [2.39–11]); poor quality of the mesorectal integrity (Incomplete + almost complete) (HR 1.81 [1.08–3.01]); the circumferential radial margin (CRM) <1 mm (HR 7.12 [3.30–15.38]); the absence or low tumor regression (TRG 1–2) (HR 1.55 [1.15–2.09]); advanced tumor T stage andpT3/T4 (HR 1.52 [1.12–2.07]); the presence of pathological lymphadenopathy after neoadjuvant treatment and ypN+ (HR 1.51 [1.08–2.10]); the presence of three or more positive nodes (HR 2.48 [1.61–3.82]); involvement of 50% or more of isolated nodes (HR 1.91 [1.16–3.17]); the ratio of positive nodes ≥0.28 (HR 2.11 [1.42–3.14]); the presence of perineural invasion (HR 2.57 [1.69–3.89]); the presence of perivascular invasion (HR 1.88 [1.22–2.91]) and the absence of adjuvant QT administration (HR 2.08 [1.54–2.82]) (Figure 2).

### 3.9. Disease-Free Survival (DFS)

Univariable analysis for overall recurrence according to the main variables influencing recurrence (Table 5). The variables that significantly influenced overall recurrence in the univariable analysis were as follows: surgical technique, postoperative complications, type of resection R0 vs. R1, radial circumferential margin, degree of tumor regression, stage ypT, stage ypN, total number of positive nodes, percentage of involved nodes, positive node ratio ≥ 0.5, positive node ratio ≥ 0.16 and existence of perineural and perivascular invasion (Figure 3).

#### 3.9.1. Local Recurrence-Free Survival (LRFS)

Univariable analysis for local recurrence according to the main variables influencing local recurrence were as follows (Table 6): anesthetic risk, toxicity to neoadjuvant radiotherapy, surgical technique, postoperative complications, type of resection R0 vs. R1, radial circumferential margin, degree of tumor regression, stage ypN, total number of positive nodes, percentage of affected nodes, positive node ratio ≥ 0.5, positive node ratio ≥ 0.25 and the existence of perineural and lymphovascular invasion (Figure 4). 

#### 3.9.2. Univariable Analysis ypN0/ypN+

With a median follow up of 7 years (P25 48 months-P75 134.8 months) the following events occurred in the series:

**Overall survival (OS) ypN0/ypN+.** Statistically significant differences in overall survival at 7 years were found between the ypN0 and ypN+ groups after analysis using the log-rank test: OS at 7 years 63.8% vs. 55.7%, respectively, (*p* = 0.016). The overall mortality rate in the ypN+ group is 1.51 times higher than in the ypN0 group. HR 1.51; CI (1.08–2.10); *p* = 0.016 (Figure 2).

**Disease-free survival (DFS) ypN0/ypN+.** Statistically significant differences in disease-free survival at 7 years were found between the ypN0 and ypN+ groups after analysis using the log-rank test; 7-year DFS in the ypN0 group is 78% and in the ypN+ group 53.8%. The overall recurrence rate in the ypN+ group is 2.46 times higher than in the ypN0 group. HR 2.46; CI (1.72–3.5) (Figure 3).

**Local recurrence-free survival (LRFS) ypN0/ypN+.** Statistically significant differences in local recurrence-free survival at 7 years were found between the ypN0 and ypN+ groups after log-rank test analysis: SLRL at 7 years in the ypN0 group is 93.6% and in the ypN+ group it is 84%. The local recurrence rate in the ypN+ group is 2.71 times higher than in the ypN0 group. HR 2.71; CI (1.46–5.06); *p* = 0.002 (Figure 4).

## 4. Discussion

In patients with locally advanced rectal cancer treated with neoadjuvant chemoradiation followed by total mesorectal excision (TME), the categorization of ypN+ tumors is relevant as they are a particular group of neoplasms that are made up of radiochemoresistant cell populations, which are related to an adverse prognosis. In early studies exploring protracted pelvic fractionated preoperative irradiation and fluopirimidins as potentiating agents, Sauer et al. found that all patients who developed recurrence had post-treatment lymph node involvement (ypN+) [7]. In this regard, the Italian group of Capirici et al. [8] performed an extensive analysis of 566 complete responders (ypCR) patients and observed a lack of correlation between the number of nodes collected and the prognosis of ypCR patients. The Japanese group of Beppu et al. [9] investigated the requirements for defining complete nodal regression and the criteria were TRG 3 response of the primary tumor and node size <6 mm. Quah et al. [10] prospectively analyzed 342 rectal cancer patients treated with neoadjuvant therapy at Memorial Sloan-Kettering Cancer Center and concluded that, although tumor regression is a strong predictor of disease-free survival, this event is most accurately predicted by pathological stage and ypTN. Onaitis et al. [11] analyzed 141 patients of similar characteristics and observed that tumor pathological stage ypT had no effect on oncological outcomes; however, nodal stage ypN was confirmed as the only independent factor for recurrence and survival. A recent study performed to identify the true percentage of patients eligible for organ-sparing approaches and conservative surgery reported patients with ypN0 was 62.7%; assuming that 37.2% survived, an organ-sparing approach would have not been oncologically correct, with parameters associated with poorer disease-free survival being ypN+ (*p* = 0.019), higher lymph node ratio (*p* = 0.001) and higher stage (*p* = 0.038) [12].

The introduction of oxaliplatin-based induction chemotherapy in the neoadjuvant segment of treatment [3], the systematic use of fluoprimidine potentiation of pelvic irradiation [4], the addition of a intraoperative presacral electron boost [5] and the progressive introduction of adjuvant oxaliplatin-based chemotherapy component in high-risk patients for recurrence are local and systemic intensification approaches developed in a prolonged period of time (20 years prospectively registered experience) in an expert institution on rectal cancer multimodal treatment. The data generated allows us to evaluate the impact of abundant clinical, pathological and treatment variables involved in the outcome of rectal cancer patients in such a long-time period. To better understand the dominant feature for outcome adversity in rectal cancer control, metastatic, persistent and resistant nodal disease as a measurable biological disease category has been analyzed in depth.

The importance of pathological nodal stage ypN as a post-neoadjuvant factor has been reported by multiple groups. Rödel et al. [13], identified that ypT, ypN, yp stage, TRG and histological type were associated in the univariable analysis with SLE, and ypN was the strongest prognostic factor in the multivariable analysis. The Korean group of Kim et al.l, conducted a rigorous study [14] to determine the factors affecting survival after neoadjuvant treatment. They retrospectively analyzed 114 patients and observed in univariable analysis that the factors significantly affecting survival were: CEA level and stage ypT and ypN. The 5-year survival was 42.3% in the ypN+ group and 73.8% in the ypN0 group (*p* = 0.0004). In multivariable analysis, ypN stage was confirmed as an independent prognostic factor for recurrence and overall survival. In a subsequent update [15], in this case of 420 patients, the results revealed in multivariable analysis that ypT stage, TRG and ypN were independent prognostic factors for SLE, with nodal stage being the prognostic factor with the best ability to discriminate 5-year SLE, which was 84.8% for ypN0, 68% in ypN1 and 27% for ypN2 (*p* < 0.001).

Rather than assessing ypT or ypN separately, some studies have evaluated ypTN as a single predictor of long-term oncological outcome [16,17,18]. Kuo et al. [19] reported that ypTN correlated with SLE. Moreover, Wei et al. concluded that ypTNM stage combined with TRG can assess oncological prognosis more accurately.

In our study, in the univariable analysis, post-neoadjuvant pathological lymph node stage (ypN) significantly influenced the three oncological events (OS, SLE, SLRL): In overall survival, the overall mortality rate in the ypN+ group was observed to be 1.51 times higher than in the ypN0 group; In disease-free survival, the overall recurrence rate in the ypN+ group was evidenced to be 2.46 times higher than in the ypN0 group and in local recurrence-free survival, the local recurrence rate in the ypN+ group was found to be 2.71 times higher than in the ypN0 group.

Kim et al. performed a retrospective study [20] in which they analyzed 421 patients treated with neoadjuvant followed by TME in four groups according to GLC quartiles: GLC1: ≤1; GLC2: ≤0.2; GLC3: <≤0.4 and GLC4: >0.4. Univariable analysis showed that 5-year OS decreased with increasing GLC: GLC1: 89%, GLC2: 67%, GLC3: 64% and GLC4: 50% (*p* < 0.001). In multivariable analysis, GLC was also shown to be a statistically significant independent prognostic factor for GLC3: ≤0.4 and GLC4: >0.4 (HR2.4 and 3.7, respectively) (*p* = 0.005). Dekker et al. [21] conducted a study including 605 patients categorized into four prognostic groups according to GLC quartiles and demonstrated that GLC was statistically significantly an independent risk factor for OS and EFS. Furthermore, they stipulated that the best GLC cut-off point to identify high-risk patients was 0.60; when analyzing the two groups GLC < 0.60 vs. GLC ≥ 0.6, they observed that 5-year OS was significantly better in the GLC < 0.60 OS group 61 vs. 32% (HR 2.45 *p* < 0.001). The same was true for SLRL (HR 1.65 *p* < 0.001).

### 4.1. Impact of the Nodal Quotient

In our series, the best cut-off point for each of the study events was estimated using the ROC curve (Figure 5) and the Youden index: GLC 28% for OS, GLC16% for SLE and GLC25% for SLRL. However, as these figures are not easily recognizable, the same study was also performed using 50% as the cut-off point, a standard and easily remembered figure in routine clinical practice, and the results were consistent for both cut-off points: Univariate analysis revealed that a high GLC (either >50% or calculated by the Youden Index) is statistically significantly associated with worse outcomes in overall survival, disease-free survival and local relapse-free survival. The overall mortality rate was twice as high for both GC ≥ 28% (HR2.11: *p* < 0.001) and GC ≥ 50% (HR 1.91; *p* = 0.012). The overall recurrence rate was threefold for GC ≥ 16% (HR 3.61: *p* < 0.001) and twofold for GC ≥ 50% (HR 2.35; *p* = 0.002). The local recurrence rate was four-fold higher for GC ≥ 25% (HR 4.13: *p* < 0.001) and three-fold higher for GC ≥ 50% (HR 3.00; *p* = 0.014). However, the relationship between metastatic lymph nodes and poor oncologic outcomes in patients treated with preoperative CRT followed by radical surgery implies that the value of the lymph node ratio after CRT is still controversial and under study, although it is true that the data so far show that a high positive ratio is associated with poor survival.

### 4.2. Total Number of Nodes Assessed

There is a consensus that the greater the number of lymph nodes collected and analyzed, the greater the likelihood of finding metastatic lymph nodes, which allows for correct staging of the disease and consequently for planning the most appropriate adjuvant treatment and the best approximation of the patients’ long-term prognosis [22]. Conversely, some authors have published that the absence of lymph nodes is associated with pathological character and favorable oncological outcomes and should reflect the best response to CRT rather than suboptimal radicality [23]. Habr-Gama et al. [24] published 281 patients after neoadjuvant radiochemotherapy prior to radical surgery and classified into three groups: ypNx, ypN0 and ypN+. The 5-year disease-free survival analysis showed significantly better results in the group with no lymph node findings (ypNx) compared to the other two groups (ypN0 and ypN+) (74% vs. 59% vs. 30%, *p* > 0.001). However, the oncological impact of “less than 12 lymph nodes” in post-neoadjuvant patients is still uncertain and it is unknown whether the prognosis of the number of metastatic lymph nodes differs between patients with only a small number of nodes harvested compared to those with a larger number. In our series, only 25.6% of patients had 12 or more lymph nodes collected. In the univariable analysis, no statistically significant differences were observed between the two groups (<12 or ≥12 nodes) for any of the events.

### 4.3. Total Number of Metastatic Nodes

In the univariable analysis of our series, we observed that patients with ≥3 positive nodes have worse prognosis, OS (HR 2.48 *p* < 0.001) and SLE (HR 2.35 *p* = 0.002), compared to those with <2 affected nodes, with a much higher local recurrence rate, four times higher in patients with 3 or more positive nodes, SLRL (HR4.13 *p* < 0.001). Considering that the ypN stage is the most influential and discriminative independent prognostic factor [15], it can be inferred that ypN+ patients are selected as candidates to receive a “tailor made” or different adjuvant treatment with respect to those who have not shown radio chemoresistance (ypN0). In this sense, different cut-off values for GLC, discriminating between low and high-risk patients (even in patients with less than 12 nodes analyzed) have been proposed as a tool to help select patients who could benefit from risk-adapted adjuvant treatment. The value of adjuvant chemotherapy based on pathological nodal stage ypN0 or ypN+ has been suggested as unnecessary in ypN patients regardless of ypT stage [25,26,27,28]. It is unquestionable that ypN+ patients have neoplasms with a more aggressive tumor biology than ypN0. In our experience, ypN+ patients have a higher ypT3- T4 rate (75.9% vs. 35.8%, respectively, *p* < 0.001) than ypN0, more patients with no tumor downstaging (59.6% vs. 25.5% *p* < 0.001), a higher rate of patients with little tumor regression (TRG 1–2) (84% vs. 42.2% *p* < 0.001), and a higher proportion of patients with little tumor regression (TRG 1–2) (84% vs. 42.2% *p* < 0.001) and a higher proportion of ypT3- T4 patients with tumor regression (84% vs. 42.2% *p* < 0.001) and a higher proportion of patients with perineural (48.7% vs. 8.1% *p* < 0.001) and lymphovascular invasion (54.5% vs. 7.6% *p* < 0.001), all of which were statistically significantly associated in the univariable analysis with worse oncological outcomes (OS, EFS and SLRL).

Some original components of information obtained in the present study are in relation to the specific study of biological features available in the pathology exam and the long-term treatment effects follow up. As biological features it is of interest to observe the significant correlation of presence of chemoradiation resistant cells in the process of migration (perineural and lymphovascular invasion) and the evidence of intranodal metastatic colonization and persistence. In terms of the local effects of treatment on the ypN disease category, the effect of neoadjuvant oxaliplatin-based chemotherapy of significantly decreasing the incidence of ypN+ surgical specimens has been seldom described. Local treatment was intensified by the administration of an intraoperative pelvic electron beam boost (80% of patients treated); no differences in local recurrence among ypN groups were observed. The adversity of ypN+ biology was also significant related to surgical local factors such as R status and circumferential margin involvement. To decrease a 26% local recurrence rate at 7 years in the ypN+ patients after local and systemic intensified treatment programs is not a minor challenge. In terms of radiotherapy opportunities proton therapy is a valid available option. The dosimetry of protons can be well adapted to the posterior pelvic space and this boosting decision can be taken once the ypN+ status postoperatively [29]. Special protection of the intrapelvic sensitive tissues (bowell, bladder, etc.) can be implemented by filing the posterior pelvic area with spacers [30]. Omentum is the recommended material to use as spacer for abdominal proton beam [31] due to the uncertainties for anatomical migration of other type of devices [32].

On the other hand, it is also suggested that ypN+ patients with higher nodal tumor burden, i.e., with high GLC, should receive a personalized or risk-adapted adjuvant QT, different from conventional and more intense than the rest of the patients [25], as a discriminatory value of ypN+ is identified as an indicator of radiochemoresistance in migrant rectal cancer population, which has a lower adverse oncological impact on disease control. Between the two extremes there is an intermediate group of patients classified as responders to preoperative CRT and sensitive to oxaliplatin induction therapy (ypN0, TRG3-4 but who unlike the first group have other poor prognostic factors) who, as Calvo et al. [33] point out in an original study, would benefit from adjuvant chemotherapy with oxaliplatin if a sufficient cumulative dose (>5 cycles) is achieved. However, thus far there have been no prospective randomized studies comparing the administration or not of adjuvant QT according to ypN+/ypN− patient groups or within ypN0 patients according to whether they have other factors associated with more or less aggressive tumor biology. The findings of the present study indicate the need for innovative randomization in ypN+ patients categories. There are no published studies comparing the type of treatment regimen within ypN+ patients according to whether they have a low GLC/high GLC. Therefore, innovation in clinical practice still requires clinical trials to define targeted therapies adjusted for risk and oncological prognosis in patients with metastatic, refractory and persistent nodal disease after long-course neoadjuvant therapy. The potential of targeted therapy depends on the correlation with molecular expression in the nodal resistant tissue. Nevertheless, our group has investigated molecular imaging and molecular-pathological response after neoadjuvant treatment in rectal cancer and the correlations were multifactorial including elements of patients’ viral exposure [34].

### 4.4. Limitations

This study has several limitations which potentially can bias the interpretations of results. It has the known limitations of observational descriptive studies, collecting data in a prolonged period in a multidisciplinary treatment oncological program. It is an unicentric experience with the bias introduced by the institutional performance of multi-specialist teams based on changing technological pharmacological resources. On the other hand, the plethora of relevant medical variables, the systematic data registration along a prolonged period and the homogeneity of a single-institution medical and surgical practice standards are factor to partially balance the intrinsic limitations of the methodology employed.

## 5. Conclusions

The presence of nodal metastases (ypN+) after neoadjuvant therapy containing long-course pelvic irradiation severely impacts long-term outcome of patients with locally advanced rectal cancer and correlates with multiple clinical and therapeutic variable metrics. The data generated in a 20-year period, including elements of treatment intensifications, both for local tumor control promotion (intraoperative electron boost) and to decrease systemic risk (neo and adjuvant chemotherapy oxaliplatin-based components), indicates that resistant metastatic nodal disease in locally advanced rectal cancer patients should be considered a subcategory for specific trial design. The remaining systemic treatment component in multimodal management should be risk-adapted to the ypN status (not to insist in agents used in the neoadjuvant segment if proven resistance is present). Additional intensification for pelvic tumor control promotion in ypN+ patients can be further explored using a combination of strategies including the use of spacers and proton therapy [35].

## Figures and Tables

**Figure 1 cancers-15-04591-f001:**
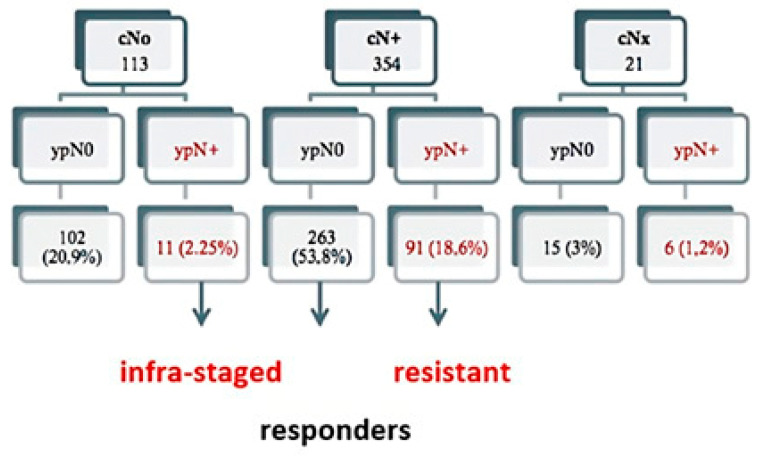
Correlations and distribution among clinical and pathological nodal stages in terms of disease categories.

**Figure 2 cancers-15-04591-f002:**
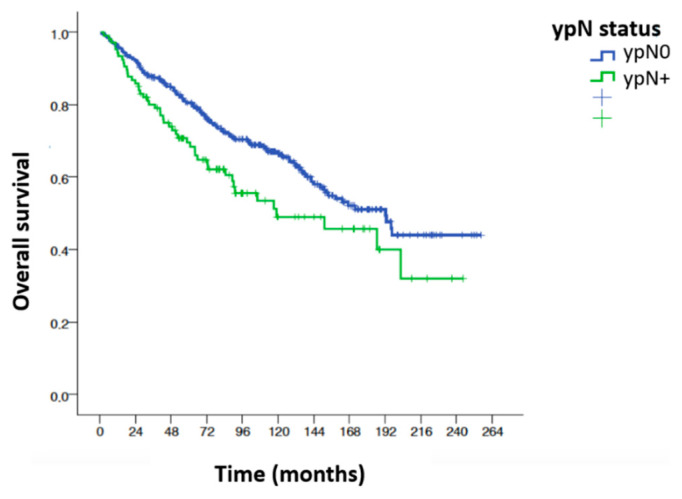
Overall survival and ypN status.

**Figure 3 cancers-15-04591-f003:**
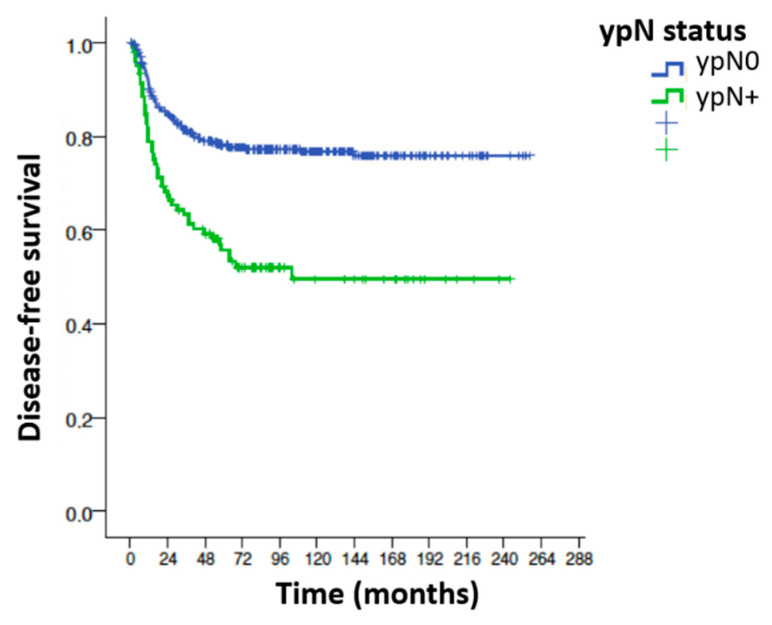
Disease-free survival and ypN status.

**Figure 4 cancers-15-04591-f004:**
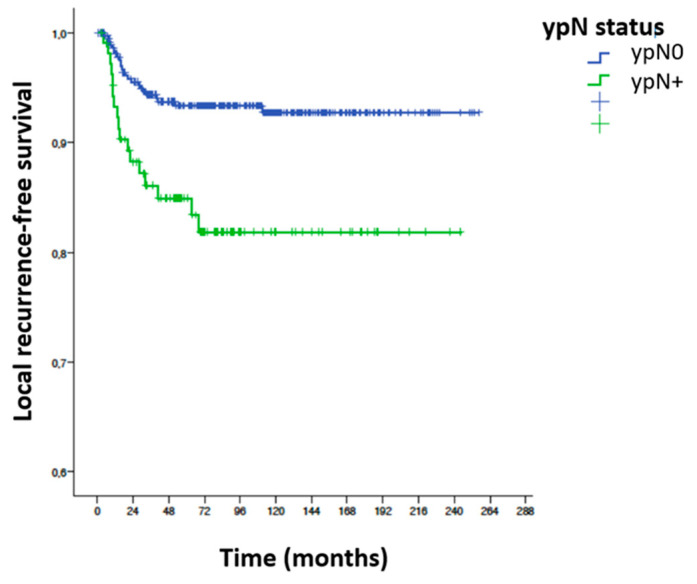
Local recurrence-free survival and ypN status.

**Figure 5 cancers-15-04591-f005:**
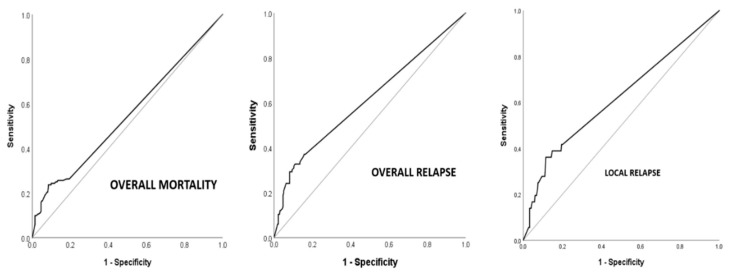
ROC curves analysis for nodal quotients in relation to patterns of cancer recurrence and mortality.

**Table 1 cancers-15-04591-t001:** Demographics variables.

Variable	Categories	YpN0 n (%)	ypN+ n (%)	Total n (%)	*p* Value
Sex	Male Female	233 (61.3) 147 (0.7)	64 (59.3) 44 (40.7)	297 (60.9) 191 (39.1)	0.699
Age	<65 ≥65	161 (42.4) 219 (57.6)	61 (56.5) 47 (43.5)	222 (45, 5) 266 (54.5)	0.009
ASA	1–2 3–4	133 (76.9) 40 (23.1)	53 (88.3) 7 (11.7)	186 (79.8) 47 (20.2)	0.057
Previous Surgery	No Yes	276 (73.4) 100 (26.6)	80 (74.8) 27 (25.2)	356 (73.7) 127 (26.3)	0.778

**Table 2 cancers-15-04591-t002:** Pre-surgical variables.

		ypN0	ypN+	Total	
		n (%)	n (%)	n (%)	*p*
Segment	Lower Medium Upper	131 (34.5) 204 (53.7) 45 (11.8)	44 (40.7) 53 (49.1) 11 (10.2)	175 (35.9) 257 (52.7) 56 (11.5)	0.482
Anal margin distance		6.93 (3.16)	6.62 (3.01)	6.86 (3.14)	0.365
Degree of differentiation	G1 G2 G3	108 (30.3) 234 (65.5) 15 (4.2)	26 (25.2) 68 (66) 9 (8.7)	134 (29.1) 302 (65.7) 24 (5.2)	0.146
cN	cN0 cN+ cNx	102 (26.8) 263 (69.2) 15 (3.9)	11 (10.2) 91 (84.3) 6 (5.6)	113 (23.2) 354 (72.5) 21 (4.3)	**0.001**
cT	cT2 cT3 cT4 cTx	26 (6.8) 288 (75.8) 59 (15.5) 7 (1.8)	3 (2.8) 76 (70.4) 25 (23.1) 4 (3.7)	29 (5.9) 364 (74.6) 84 (17.2) 11 (2.3)	0.075
Clinical Stage	II III	102 (27.9) 263 (72.1)	11 (10.8) 91 (89.2)	113 (24.2) 354 (75.8)	**<0.001**
Pre-operative RT dose	<5040 ≥5040	47 (12.4) 331 (87.6)	19 (17.6) 89 (82.4)	66 (13.6) 420 (86.4)	0.168
Full chemotherapy	No Yes	70 (20.3) 275 (79.7)	21 (22.3) 73 (77.7)	91 (20.7) 348 (79.3)	0.664
Toxicity RT	No Yes	67 (18.2) 302 (81.8)	21 (19.8) 85 (80.2)	88 (18.5) 387 (81.5)	0.699
Time to surgery	<8 ≥8	305 (80.5) 74 (19.5)	86 (79.6) 22 (20.4)	391 (80.3) 96 (19.7)	0.846

**Table 3 cancers-15-04591-t003:** Post-surgical variables.

		ypN0	ypN+	Total	
		n (%)	n (%)	n (%)	*p*
ypT	ypT1/T2 ypT3/T4	206 (62.2) 115 (35.8)	26 (24.1) 82 (75.9)	232 (54.1) 197 (45.9)	<0.001
No. of identified nodes	<12 ≥12	270 (78.0) 76 (22.0)	64 (61.5) 40 (38.5)	334 (74.2) 116 (25.8)	<0.001
N-downstaging	No Yes Other	102 (28.0) 262 (72) 0 (0)	91 (89.2) 0 (0) 11 (10.8)	193(41.2) 262 (56.2) 11(2.4)	<0.001
T-downstaging	No Descent 1 T Decrease 2 T Downturn 3 T Worsening	95 (25.5) 150 (40.3) 54 (14.5) 63 (16.9) 10 (2.7)	62 (59.6) 33 (31.7) 5 (4.8) 0 (0) 4 (3.8)	157 (33.0) 183 (38.4) 59 (12.4) 63 (13.2) 14 (2.9)	<0.001
Mesorectum Quality	Incomplete + almost full Full	50 (30.1) 116 (69.9)	31 (47.7) 34 (52.3)	81 (35.1) 150 (64.9)	0.012
Type of resection	R1 R0	3 (0.8) 377 (99.2)	5 (4.6) 103 (95.4)	8 (1.6) 480 (98.4)	0.015
Tumor regression grade (TRG)	3–4 (few tumor cells and fibrosis/no tumor cells, only fibrotic tissue) 1–2 (tumor dominant tissue and evidence of fibrosis)	218 (57.8) 159 (42.2)	17 (16) 89 (84)	235 (48.7) 248 (51.3)	<0.001
Circumferential resection margin (CRM)	Free Affect	376 (99.2) 3 (0.8)	102 (95.3) 5 (4.7)	478 (98.4) 8 (1.6)	0.015
Perineural invasion	No Yes	238 (91.9) 21 (8.1)	40 (51.3) 38 (48.7)	278 (82.5) 59 (17.5)	<0.001
Lymphovascular invasion	No Yes	244 (92.4) 20 (7.6)	35 (45.5) 42 (54.5)	279 (81.8) 62 (18.2)	<0.001
Adjuvant chemotherapy	No Yes	102 (26.8) 278 (73.2)	16 (14.8) 92 (85.2)	118 (24.2) 370 (75.8)	0.010
Type adjuvant chemotherapy	Mayo Folfox Xeloda Folfiri Xelox Other	99 (26.3) 141 (37.4) 17 (4.5) 1 (0.3) 15 (4.0) 8 (2.1)	25 (23.4) 49 (45.8) 5 (4.7) 0 (0.0) 9 (8.4) 4 (3.7)	124 (25.6) 190 (39.3) 22 (4.5) 1 (0.2) 24 (2.9) 12 (2.5)	<0.001

**Table 4 cancers-15-04591-t004:** Multivariable analysis by log-rank test and invariant Cox regression of overall survival as a function of the main prognostic factors. (Abbreviations: Inf: inferior; sup: superior).

			Exitus	IC		
Variable		n	HR	Inf	Sup	*p*
Pre-surgical						
Age	<65	220	0.48	0.36	0.66	<0.001
	≥65	260	1			
ASA	1–2	186	1			
	3–4	45	2.11	1.15	3.88	0.016
Height Tumor	Lower	174	1.49	1.10	2.04	0.010
	Medium	251	1			
	Upper	55	1.12	0.68	1.82	0.646
Toxicity RT neoadjuvant	No	87	1			
	Yes	380	0.92	0.63	1.33	0.648
Surgical						
Surgical Technique	Sphincter-preserving	329	1			
	AAP	145	1.75	1.30	2.35	<0.001
Laparoscopy	No	352	1			
	Yes	135	0.38	0.23	0.644	<0.001
Total mesorectal excision (TME)	No	55	1.05	0.66	1.68	0.80
	Yes	425	1			
Resection R0 vs. R1	No	9	5.13	2.39	11.00	<0.001
	Yes	471	1			
Complication	No	286	1			
	Yes	169	1.79	1.311	2.442	<0.001
Post-surgical						
Quality Mesorectum	Incomplete + almost complete	79	1.81	1.08	3.01	0.023
	Complete	148	1			
Mesorectal circumferential margin	Free	470	1			
	Involved	8	7.12	3.30	15.38	<0.001
Tumor regression grade	3–4	232	1			
	1–2	243	1.55	1.15	2.09	0.004
ypT	ypT1/T2	227	1			
	ypT3/T4	194	1.52	1.12	2.07	0.007
ypN	ypN0	373	1			
	ypN+	106	1.51	1.08	2.10	0.016
No. of isolated lymph nodes	<12	328	0.79	0.519	1.22	0.295
	≥12	113	1			
No. nodes +	0–2	438	1			
	≥3	40	2.48	1.61	3.82	<0.001
Node ratio + 28%	<0.28	382	1			
	≥0.28	53	2.11	1.42	3.14	<0.001
Node quotient + 50%	<50%	406	1			
	≥50%	29	1.91	1.16	3.17	0.012
Perineural Invasion	No	275	1			
	Yes	58	2.57	1.69	3.89	<0.001
Lymphovascular Invasion	No	276	1			
	Yes	61	1.88	1.22	2.91	0.004
Adjuvant chemotherapy	No	110	2.08	1.54	2.82	<0.001
	Yes	370	1			

**Table 5 cancers-15-04591-t005:** Multivariable analysis using log-rank test and Cox invariant regression of overall recurrence according to the main prognostic factors. (Abbreviations: Inf: inferior; sup: superior).

			R. Global	IC		
Variable		n	HR	Inf	Sup	*p*
**Pre-surgical**						
**Age**	<65	220	1.15	0.82	1.62	0.412
	≥65	260	1			
**ASA**	1–2	186	1			
	3–4	45	1.19	0.63	2.24	0.600
**Height Tumor**	Lower	174	1.19	0.82	1.71	0.353
	Medium	251	1			
	Upper	55	1.16	0.67	2.01	0.582
**Toxicity RT** **neoadjuvant**	No	87	1			
	Yes	380	0.77	0.51	1.17	0.221
**Surgical**						
**Surgical Technique**	Sphincter-preserving	329	1			
	AAP	149	1.48	1.04	2.10	**0.031**
**Laparoscopy**	No	345	1			
	Yes	135	0.77	0.51	1.16	0.216
**Total mesorectal excision (TME)**	No	55	0.99	0.547	1.55	0.762
	Yes	425	1			
**Resection R0 vs. R1**	No	9	4.35	1.91	9.89	**<0.001**
	Yes	471	1			
**Complication**	No	286	1			
	Yes	169	1.61	1.13	2.29	**0.009**
**Post-surgical**						
**Quality Mesorectum**	Incomplete + almost complete	79	1.60	0.93	2.73	0.088
	Complete	148	1			
**Mesorectal** **circumferential margin**	Free	470	1			
	Involved	8	5.72	2.51	13.05	**<0.001**
**Tumor regression grade**	3–4	232	1			
	1–2	243	2.66	1.83	3.89	**<0.001**
**ypT**	ypT1/T2	227	1			
	ypT3/T4	194	2.29	1.59	3.31	**<0.001**
**ypN**	ypN0	373	1			
	ypN+	106	2.46	1.73	3.51	**<0.001**
**No. of isolated lymph nodes**	<12	328	1.09	0.72	1.64	0.679
	≥12	113	1			
**No. nodes +**	0–2	438	1			
	≥3	40	3.27	2.08	5.14	**<0.001**
**Node ratio + 16%**	<0.16	362	1			
	≥16	73	3.061	2.07	4.51	**<0.001**
**Node quotient + 50%**	<50%	406	1			
	≥50%	29	2.35	1.37	4.05	**0.002**
**Perineural Invasion**	No	275	1			
	Yes	58	2.95	1.91	4.54	**<0.001**
**Lymphovascular** **Invasion**	No	276	1			
	Yes	61	2.73	1.78	4.19	**<0.001**
**Adjuvant** **chemotherapy**	No	110	1.26	0.85	1.87	0.252
	Yes	370	1			

**Table 6 cancers-15-04591-t006:** Multivariable analysis by log-rank test and Cox invariant regression of local recurrence according to the main prognostic factors. (Abbreviations: Inf: inferior; sup: superior).

			R. Local	IC		
Variable		n	HR	Inf	Sup	*p*
**Pre-surgical**						
**Age**	<65	220	1.03	0.56	1.88	0.931
	≥65	260	1			
**ASA**	1–2	186	1			
	3–4	45	2.76	1.00	7.61	**0.049**
**Height Tumor**	Lower	174	1.23	0.65	2.35	0.511
	Medium	251	1			
	Upper	55	0.89	0.30	2.61	0.841
**Toxicity RT** **neoadjuvant**	No	87	1			
	Yes	380	0.41	0.21	0.78	**0.007**
**Surgical**						
**Surgical Technique**	Sphincter preserving	329	1			
	AAP	149	2.39	1.28	4.44	**0.006**
**Laparoscopy**	No	345	1			
	Yes	135	0.51	0.22	1.16	0.112
**Total mesorectal** **excision**	No	55	1.22	0.43	3.42	0.704
	Yes	425	1			
**Resection R0 vs. R1**	No	9	9.72	3.45	27.41	**<0.001**
	Yes	471	1			
**Complication**	No	286	1			
	Yes	169	2.15	1.16	3.40	**0.014**
**Post-surgical**						
**Quality Mesorectum**	Incomplete + almost complete	79	0.87	0.30	2.51	0.801
	Complete	148	1			
**Mesorectal** **circumferential margin**	Free	470	1			
	Affect	8	7.69	2.35	25.12	**0.001**
**TRG**	3–4	232	1			
	1–2	243	3.16	1.55	6.45	**0.002**
**ypT**	ypT1/T2	227	1			
	ypT3/T4	194	1.58	0.83	2.99	0.162
**ypN**	ypN0	373	1			
	ypN+	106	2.71	1.46	5.06	0.002
**No. of isolated lymph nodes**	<12	328	1.03	0.48	2.21	0.931
	≥12	113	1			
**No. nodes +**	0–2	438	1			
	≥3	40	2.66	1.18	6.02	**0.018**
**Node ratio + 25%**	<0.25	376	1			
	≥0.25	59	4.13	2.09	8.17	**<0.001**
**Node quotient + 50%**	<50%	406	1			
	≥50%	29	3.00	1.25	7.21	**0.014**
**Perineural Invasion**	No	275	1			
	Yes	58	2.47	1.21	5.02	**0.013**
**Lymphovascular** **Invasion**	No	276	1			
	Yes	61	2.36	1.16	4.83	**0.018**
**Adjuvant** **chemotherapy**	No	110	1.88	0.99	3.57	0.054
	Yes	370	1			

## Data Availability

The datasets used and/or analyzed during the current study are available from the corresponding author on reasonable request.

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
