# Peer review of "Post-Chemoradiation Metastatic, Persistent and Resistant Nodes in Locally Advanced Rectal Cancer: Metrics and Their Impact on Long-Term Outcome"

_cancers, 2023, doi:10.3390/cancers15184591_

Round 1
Reviewer 1 Report
The authors wrote an observational retrospective study to evaluate long-term oncological outcome in patients with lymph node involvement after neoadiuvant therapy for locally advanced rectal cancer (ypN+).
They analyzed 487 patients undergone to surgical resection for rectal cancer from April 1995 to March 2015, 108 of which resulted ypN+.
Bearing in mind the international scientific effort to develop a tailored therapeutic approach for cancer patients, the analysis of pathological and clinical factors that can affect long-term outcomes of patients with rectal cancer appears necessary, but there are major issue to be solved:
· Tables appears to lack all unkown data (for example, in Table 1 ASA score is calculated just for 173 patients in pN0 group and 60 patients in pN+ group). All missing data have to be specified and percentage must be calculated, otherwise this limit need to be discussed;
· In the paragraph “N-downstaging” of results, appears unclear nodal status before neoadjuvant chemotherapy, please specify and discuss;
· Limitations and biases of the study are never examined.
Also, there are minor issues to complete:
· Many p values appears to be 0.000, please insert correct value (Lines 219, 274, 401, 402 and 439);
· The first sentence of “Perineural and lymphovascular invasion” paragraph seems mistyped, please correct;
· Last sentence of “pattern of progression” seems mistyped, please correct;
Finally, please consider the discussion of this observational study published in 2023 focused on the critical role of lymph node metastasis in rectal cancer’ treatment:
L Franca A et al. Lymph Node Metastasis in Extraperitoneal Rectal Cancer After Neoadjuvant Therapy: An Unsolved Problem? Anticancer Res. 2023 Jun;43(6):2813-2820. doi: 10.21873/anticanres.16450. PMID: 37247907.
There are minor issues to be solved:
- The first sentence of “Perineural and lymphovascular invasion” paragraph seems mistyped, please correct;
- Last sentence of “pattern of progression” seems mistyped, please correct;
Author Response
Please see the attachment. This document provides a point-by-point response to the reviewer's comments.

Reviewer 2 Report
The authors should be commended for the large sample size of their cohort and the plethora of relevant variables that they recorded. However, I have several major concerns:
1. Please explain abbreviations (eg sup, inf, RIVER etc in table 4), MRC etc.
2. Why did you estimate RFS and OS at 7 years and not the typically used 5 years endpoint?
3. The results of the multivariable analysis are briefly reported in the discussion but they should be reported in detail in tables and text in the results section. If Table 5 for example is one of these tables why there is no mention of "multivariable"?
4. The correct term is multivariable and not multivariate analysis.
5. Lines 474-476: "n our series, the best cut-off point for each of the study events was estimated using 475 the ROC curve and the Youden index: GLC 28% for OS, GLC16% for SLE and GLC25% 476 for SLRL."
Where is this analysis?
6. I don't see the novelty of this study. You concluded that "The presence of nodal metastases (ypN+) after neoadjuvant therapy containing 552 long-course pelvic irradiation severely impacts long-term outcome of patients with locally 553 advanced rectal cancer." This is intuitive and has been shown repeatedly by several groups as you also mention in the discussion. Thus, what is the novel contribution of your study?
Moderate editing is needed.
Author Response

(The authors gave the same response as above.)

Reviewer 3 Report
Dr. Calvo and colleagues' study showed that the presence of nodal metastases (ypN+) in locally advanced rectal cancer patients treated with neoadjuvant chemoradiation followed by TME severely impacts long-term outcomes. The study emphasized the significance of pathological nodal stage (ypN) as a prognostic factor, with higher ypN stages associated with poorer survival rates. The authors did an excellent job in providing comprehensive insights into the impact of nodal status on disease control.
Dr. Calvo and the team should be congratulated for their meticulous work. While the authors did an excellent job, there are some areas for improvements:
For minor revision, authors might:
- Provide more details on the patient selection process and criteria for inclusion in the study.
- Clarify the rationale for choosing specific adjuvant chemotherapy regimens for different subgroups of patients.
- Include a more comprehensive discussion on potential side effects and complications associated with the proposed combination of spacers and proton therapy.
While the authors mention the potential of combining spacers and proton therapy for pelvic tumor control, further elaboration and discussion on this topic would be beneficial. Additionally, incorporating the conclusion's key points into the discussion would strengthen the overall coherence of the study.
Major: It would be valuable to discuss the potential of a prospective randomized studies comparing different treatment regimens based on nodal status (ypN+/ypN-) and nodal tumor burden to define targeted therapies for risk-adapted adjuvant treatments.
Authors might:
- Address potential confounding factors, such as patient comorbidities and previous treatments, to strengthen the study's internal validity.
- Consider a more in-depth analysis of treatment response based on molecular subtypes of rectal cancer to identify potential targeted therapeutic approaches.
Overall, I thoroughly enjoyed reading their manuscript. The study presents promising results, but addressing these minor and major revision points would enhance the study's impact and validity in the field of rectal cancer management.
Round 2
Reviewer 1 Report
The authors have made the appropriate revisions and the article can be accepted
Reviewer 2 Report
1. In a previous comment I asked the authors to make it clear how the findings of this study are novel. I believe this was not addressed in the revisions.
2. In a previous comment, I asked the authors where this analysis is "in our series, the best cut-off point for each of the study events was estimated using 475 the ROC curve and the Youden index: GLC 28% for OS, GLC16% for SLE and GLC25% 476 for SLRL." I believe they did not add the relevant information in methods and results section, for example, how did they come up with the GLC cut offs, where the ROC curve is etc.
moderate editing needed.
Round 3
Reviewer 2 Report
The authors have revised more carefully.
The manuscript needs extensive editing.
Author Response
The authors are very grateful for the various contributions proposed to make the article more solid and consistent.